# Social Innovation, Employee Value Cocreation, and Organizational Citizenship Behavior in a Sport-Related Social Enterprise: Mediating Effect of Corporate Social Responsibility

**Chen-Yueh Chen, Ya-Lun Chou * and Chun-Shih Lee**

Department of Recreation and Leisure Industry Management, National Taiwan Sport University, Taoyuan 333325, Taiwan; chenchenyueh@ntsu.edu.tw (C.-Y.C.); 1081703@ntsu.edu.tw (C.-S.L.)
* Correspondence: 1070818@ntsu.edu.tw

**Abstract:** This stakeholder theory-based study explored the mediating role of employee attitudes regarding corporate social responsibility (CSR) among perceptions of social innovation (SI), value cocreation (VCC), and organizational citizenship behavior (OCB) in a sport social enterprise context. Eighty-three employees in a Taiwanese social enterprise were recruited using random sampling. A self-administered online survey was conducted for the collection of data, which were examined using linear regression analysis. The results indicated that employee attitudes regarding CSR fully mediated the relationship among perceived SI and OCB. Additionally, the attitude toward CSR was found to partially mediate the relationship between perceptions of SI and VCC. Employees' attitudes toward CSR play a critical role in increasing their VCC and OCB in addition to their perception of an organization's SI. Meaningful theoretical and practical implications were revealed.

**Keywords:** corporate social responsibility; social innovation; value co-creation; organizational citizenship behavior

## 1. Introduction

Value cocreation (VCC), which refers to the notion that services require the participation of both service producers and users [1], has received increasing attention in disciplines, such as management, marketing, and sports [2–4]. Specifically, VCC is depicted as a joint process wherein stakeholders cocreate commercial value [5]. Most literature associated with VCC in sports has focused on the consumer's role in the process of VCC in commercialized contexts such as professional and participatory sports [1,6,7], which emphasize the importance of external stakeholders (i.e., consumers or customers). However, [4] argued that a successful VCC strategy should involve sound understandings on the part of both internal and external stakeholders to avoid co-destruction, underlining the importance of internal (i.e., employees) and external stakeholders. Sport social enterprise VCC from the employee perspective remains underexplored; accordingly, research examining such a perspective is warranted.

Organizational citizenship behavior (OCB) is defined as "an individual behavior that is discretionary, not directly or explicitly recognized by the formal reward system, and that in the aggregate promotes the effective functioning of the organization" [8]. In other words, OCB refers to extra work behaviors required for superior performance [9]. Although OCB was reported in commercial sport settings [9], OCB may be equally crucial in nonprofit sport organizations. Studies on OCB have revealed the critical role it plays in nonprofit organizations [10], which are ostensibly cognizant of OCB's benefits. Sport social enterprises are, by nature, nonprofit organizations. Accordingly, exploring the role of OCB in sport social enterprise is imperative.

Social entrepreneurship has been receiving increasing attention in sports management [11]. Social entrepreneurship is viewed as a means of creating and satisfying social

values and missions through innovative methods or actions [12]. Sports is seen as an avenue for inspiring social entrepreneurship [13]. Furthermore, ref. [14] argued that social innovation (SI) is a critical element of social entrepreneurship in the sports context. Ref. [15] defined SI as "a novel solution to a social problem that is more effective, efficient, sustainable, or just than existing solutions and for which the value created accrues primarily to society as a whole rather than private individuals." Similarly, SI is depicted as the implementation of new or improved methods of promoting social change [16]. Literature regarding SI in the sports context has examined the determinants of SI in sports clubs [17], external stakeholders' roles in shaping social innovation in sports [18], and the antecedents and outcomes of SI [19]. Although SI in sports has received academic attention, empirical analyses of SI in sports have remained limited. Therefore, investigating SI in sport social enterprises is vital.

Corporate social responsibility (CSR) in sports contexts has been extensively examined. Most CSR literature has explored the topic from the consumer perspective [20–24]. Other CSR literature has investigated employee perceptions in business organizations [25,26]. The authors of [20] noted that CSR plays a crucial role in nonprofit organizations. Sport social enterprises are nonprofit organizations and may also need to consider employee perceptions toward CSR adopted by the enterprises. Nevertheless, studies examining the moderating mechanism of employee attitudes toward CSR in relation to SI and VCC remain scarce. Therefore, investigating employee attitudes toward CSR in sport social enterprises by exploring its moderating role between SI, VCC, and OCB is crucial.

The organization of this article was addressed as follows. First, the theoretical framework used in the study was presented, followed by reasoning on development of proposed hypotheses. Additionally, issues associated with research methods were reported. Empirical analysis along with its results as well as theoretical and practical implications were emphasized. Finally, limitations and conclusion of the study were articulated.

## 2. Literature Review

### 2.1. Stakeholder Theory

According to [27], proponents of the stakeholder approach argue that stakeholders are individuals directly or indirectly influencing or being influenced by the organization's activities and decision making. Specifically, the stakeholder approach involves a recognition that an organization has numerous stakeholders, each of which participates in and is influenced by the organization's performance [28]. From the CSR perspective, employees are a primary organizational stakeholder [29]. Employees are as critical to organizational performance in nonprofit organizations as they are in for-profit organizations. Empirical analysis indicates that internal CSR activities influence employees' social performance [30]. In the context of current study, a sport social enterprise contributes to society by providing sports programs to communities while simultaneously sustaining itself financially. Similar to for-profit organizations, sport social enterprises adopt policies or strategic steps that may affect employees' attitudes toward the organization and, in turn, their behavior. Moreover, CSR affects VCC and OCB from a stakeholder perspective [31,32]. A significant positive correlation between CSR and innovation investment was also reported [33], which suggests that the perceived level of innovation adopted by the organization shapes employees' attitudes toward CSR. Therefore, investigation using the research model in the current study is appropriate.

### 2.2. SI and CSR Attitudes

Innovation was demonstrated to be correlated with CSR. For instance, innovation capacity is an organizational element crucial to CSR implementation that should be combined with human resources [34]. More specifically, a perception of innovation was inferred to be related to employee attitudes toward an organization's CSR approach. Furthermore, CSR and technological innovation investment are reportedly correlated [33]. In the sport social enterprise context, SI can yield social changes through proactive social initiatives

that practically apply new ideas [17]. When employees perceive that SI is adopted by a sport social enterprise to serve the society, they may tend to be more positive about the CSR adopted by that enterprise. Accordingly, Hypothesis 1 (H1) was developed.

**H1.** *Employees' perception of SI would positively predict their attitudes toward the organization's CSR.*

### 2.3. SI and VCC

An established relationship exists between SI and VCC in the sport social enterprise context. Qualitative research conducted on sport for development and peace leaders suggested that SI relates to a reciprocal process for cocreating new or improved means of effecting social changes in sports [16]. Additionally, SI is a significant predictor of organizational performance [19]. Another empirical study indicated that SI and VCC are correlated [35]. Similarly, VCC can be derived through the development of SI [36]. SI is inferred to be a predictor of VCC, and consequently Hypothesis 2 (H2) was developed.

**H2.** *Employees' perceptions of SI would positively predict VCC.*

### 2.4. SI and OCB

In a sports context, SI promises positive social changes through innovative sports initiatives. When sport social enterprises consistently engage in SI-related actions, SI thinking may be incorporated into employees' daily routines and gradually become a part of the organizational culture. Empirical evidence indicates that organizational culture is positively correlated with OCB [9]. Additionally, empirical analysis suggests that OCB can be stimulated by altruistic motivations in a hotel industry context [37]. SI in sport social enterprises can be conceptually regarded as an expression of altruistic motivations because it involves a wish to create positive social changes. Therefore, Hypothesis 3 (H3) was formulated.

**H3.** *Employees' perceptions of SI would positively predict OCB.*

### 2.5. CSR and VCC

The correlation between CSR and VCC can be demonstrated. Specifically, VCC can occur through CSR activities from the perspective of stakeholders [38]. A study examining CSR communication reported that social media communications should include opportunities for consumers to cocreate value with the relevant brands [39]. Although social enterprise employees are internal stakeholders, empirical evidence regarding VCC from a consumer (i.e., external stakeholder) perspective may be conceptually and equivalently applied. Furthermore, researchers have emphasized three primary drivers of multistakeholder VCC: trust, inclusiveness, and openness [40]. Employees' trust in the CSR adopted by sport social enterprises is a critical factor influencing their behavior. Hence, employee attitudes toward the CSR adopted by sport social enterprises may be correlated with VCC, which yielded the proposal of Hypothesis 4 (H4).

**H4.** *Employees' attitudes toward CSR would positively predict VCC.*

### 2.6. CSR and OCB

Numerous empirical studies have demonstrated that employee perceptions of an organization's CSR can predict their OCB. Employee CSR perception was discovered to be significantly related to their OCB within the organization [41]. Additionally, a positive association between perception of CSR and OCB was revealed [42]. In a similar vein, employee perception of CSR markedly enhances their OCB [43]. Although these previous findings were indicated in business contexts, they may still be applicable in sport social enterprises, where positive attitudes toward CSR lead to greater OCB by creating more positive social changes. Consequently, Hypothesis 5 (H5) was formulated.

**H5.** *Employee attitudes about CSR would positively predict their OCB.*

## 3. Materials and Method

### 3.1. Research Setting

A sport social enterprise in Taiwan (anonymous for confidentiality reasons) was the research setting in the current study. The sport social enterprise investigated was financially supported by an internationally recognized religion. Such enterprises have promoted education, culture, the arts, sports, camping, community service, and other related activities while at the same time managing youth exchange activities worldwide to expand their fields of international friendship and promote global peace. The sport social enterprise examined in this study fit the scenarios of SI, CSR, VCC and OCB. Therefore, the sport social enterprise as the research setting in this study is reasonable and appropriate.

### 3.2. Participants and Procedure

This study was approved by the Research Ethics Committee of National Taiwan University. Through contact with supervisors of the sport social enterprise under study, a simple random sampling method was used to enroll a target sample of its employees aged over 20 years. Employees in the organization were asked whether they would like to participate in this research. The participants required 5–10 minutes to complete the questionnaire survey and submitted the questionnaire after completion. The survey participants' responses regarding SI, VCC, CSR, and OCB were collected. Valid responses to a total of 83 questionnaires were collected. The demographic variables of the participants in this study are detailed in Table 1.

**Table 1.** Demographic data of the participants ($N$ = 83).

| Variable | | Frequency | Percent (%) |
|---|---|---|---|
| Gender | Male | 24 | 28.9 |
| | Female | 59 | 71.1 |
| Age (years) | 21–30 | 25 | 30.1 |
| | 31–40 | 20 | 24.1 |
| | 41–50 | 27 | 32.5 |
| | ≥51 | 11 | 13.3 |
| Education level | Bachelor's degree | 70 | 84.3 |
| | Master's degree | 13 | 15.7 |

### 3.3. Measurement

The data measured in this study were demographic variables, SI, and attitudes toward CSR, VCC, and OCB. The demographic variables comprised gender, age, and education level. The SI scale was adapted from the work of [44]. In addition, the measure of attitudes toward CSR was adapted from the study of [45]. In addition, the VCC scale was adapted from that of [46], and finally, the OCB scale was measured using the scale proposed by [47]. All measurements were made on a 7-point Likert scale, with 1 representing strongly disagree and 7 representing strongly agree. Exploratory factor analysis (EFA) suggested satisfactory construct validity, with the eigenvalues for all the constructs exceeding 1 and the explained variance of 66%–72%. The internal consistency of the constructs was 0.89–0.93, which indicates satisfactory reliability. Table 2 describes the measurement scales and items used in the study.

### 3.4. Data Analysis

Descriptive statistics were calculated to present the characteristics of demographic variables and all questionnaire items. The small sample size in this study limited the use of confirmatory factor analysis (CFA) to examine construct validity and structural equation modeling to explore the path coefficients among the constructs under study. Instead, EFA was used to examine the construct validity of the measurement scales used in this study. Internal consistency was investigated by calculating Cronbach's alpha

coefficients. Finally, all of the proposed hypotheses in the study were investigated with linear regression analysis.

**Table 2.** Descriptive statistics for constructs and items ($N$ = 83).

| Construct/Item | M | SD | FL |
|---|---|---|---|
| Attitudes toward CSR ($\alpha$ = 0.922, VE = 0.72, EV = 5.029) | | | |
| 1. I am aware of the social programs of XXX. | 5.77 | 1.29 | 0.78 |
| 2. I know of the good things my favorite XXX does for the community. | 5.98 | 1.12 | 0.89 |
| 3. I believe XXX to be a socially responsible organization. | 6.21 | 0.95 | 0.86 |
| 4. I am aware of the programs of XXX that benefit the community. | 6.06 | 1.07 | 0.92 |
| 5. Part of the reason I like XXX is because of what they do for the community. | 6.19 | 0.95 | 0.88 |
| 6. One of the reasons I speak positively about XXX is because of what they do for the community. | 6.19 | 0.96 | 0.92 |
| 7. I buy merchandise from XXX partly because I believe they are a socially responsible organization. | 5.56 | 1.41 | 0.67 |
| SI ($\alpha$ = 0.889, VE = 0.70, EV = 3.478). | | | |
| 1. In comparison with other public service providers, our institution has introduced more innovative services. | 5.40 | 1.12 | 0.89 |
| 2. We develop products or services that meet the needs of our citizens more effectively than any other service currently available. | 5.53 | 1.15 | 0.86 |
| 3. In comparison with other public service providers, our institution introduces new services into the market more quickly. | 5.25 | 1.22 | 0.88 |
| 4. In comparison with other public service providers, our institution has a higher success rate with new service launches. | 5.48 | 1.14 | 0.85 |
| 5. Our institution is able to change or modify our current service approaches to meet the special requirements of our citizens. | 5.77 | 1.06 | 0.66 |
| VCC ($\alpha$ = 0.910, VE = 0.70, EV = 4.224). | | | |
| 1. I will tell XXX to let them know how to meet my needs better. | 5.78 | 1.07 | 0.87 |
| 2. I will tell XXX how to improve services when I have new ideas. | 5.68 | 1.09 | 0.88 |
| 3. I will tell XXX about service issues so that they can improve. | 5.75 | 1.04 | 0.87 |
| 4. I am willing to notify XXX about a problem even if the problem does not affect me. | 5.30 | 1.27 | 0.86 |
| 5. I will tell XXX if XXX provides good service to me. | 6.01 | 0.99 | 0.77 |
| 6. Even if a price error will benefit me, I will still notify XXX. | 5.30 | 1.41 | 0.79 |
| OCB ($\alpha$ = 0.901, VE = 0.66, EV = 5.241). | | | |
| 1. I frequently volunteer to do things without being asked. | 5.74 | 1.17 | 0.82 |
| 2. I often take time away from my job to help others with their work without asking for a reward. | 5.87 | 1.04 | 0.91 |
| 3. Sometimes I will coast during part of the work day when there is little work to do rather than trying to find new work (reverse coded). | 5.77 | 1.08 | 0.86 |
| 4. If possible, I take additional unauthorized breaks (reverse coded). | 5.73 | 1.25 | 0.80 |
| 5. I exert considerable effort at work. | 6.20 | 0.76 | 0.75 |
| 6. I often try to help fellow employees so they will become more productive. | 6.02 | 0.89 | 0.90 |
| 7. When possible, I take longer lunches or breaks than allowed (reverse coded). | 5.21 | 1.59 | 0.47 |
| 8. I often help others at work who have a heavy workload without being asked to do so. | 5.91 | 1.05 | 0.86 |

Note. $\alpha$: Cronbach alpha coefficient; VE: variance explained; EV: eigenvalue; M: mean; SD: standard deviation; FL: factor loading; XXX: the sport social enterprise under study.

## 4. Results

A series of linear regression analyses were performed to investigate the hypotheses proposed in the study. The results of these analyses indicated that perceptions of SI positively predicted employee attitudes toward CSR ($\beta$H1 = 0.494, t = 5.119, $p < 0.01$) and VCC ($\beta$H2 = 0.486, t = 6.241, $p < 0.01$), thus supporting H1 and H2. However, perceptions of SI did not significantly predict OCB ($\beta$H3 = 0.068, t = 0.838, $p = 0.414$), suggesting that H3 was not supported. Nevertheless, employee attitudes toward CSR positively predicted VCC ($\beta$H4 = 0.330, t = 3.636, $p < 0.01$) and OCB ($\beta$H5 = 0.741, t = 9.144, $p < 0.01$), implying that H4 and H5 were supported. Furthermore, the findings indicated that employees' attitudes toward CSR partially mediated the relationship between SI and VCC and fully mediated the relationship between SI and OCB. Table 3 presents the results of the linear regression analyses, and Figure 1 depicts the path relationship among the constructs.

**Table 3.** Results of hypothesis testing ($N = 83$).

| Hypothesized Paths | B | S.E. | $\beta$ | t | Result |
|---|---|---|---|---|---|
| H1: SI→CSR | 0.482 | 0.094 | 0.494 | 5.119 * | Supported |
| H2: SI→VCC | 0.491 | 0.092 | 0.486 | 5.355 * | Supported |
| H3: SI→OCB | 0.062 | 0.074 | 0.068 | 0.838 | Not Supported |
| H4: CSR→VCC | 0.342 | 0.094 | 0.330 | 3.636 * | Supported |
| H5: CSR→OCB | 0.695 | 0.076 | 0.741 | 9.144 * | Supported |

Note. * $p < 0.05$. B: unstandardized regression coefficient; SE: standard error; $\beta$: standardized regression coefficient; t: t-value; SI: social innovation; CSR: attitudes toward corporate social responsibility; VCC: value co-creation; OCB: organizational citizenship behavior.

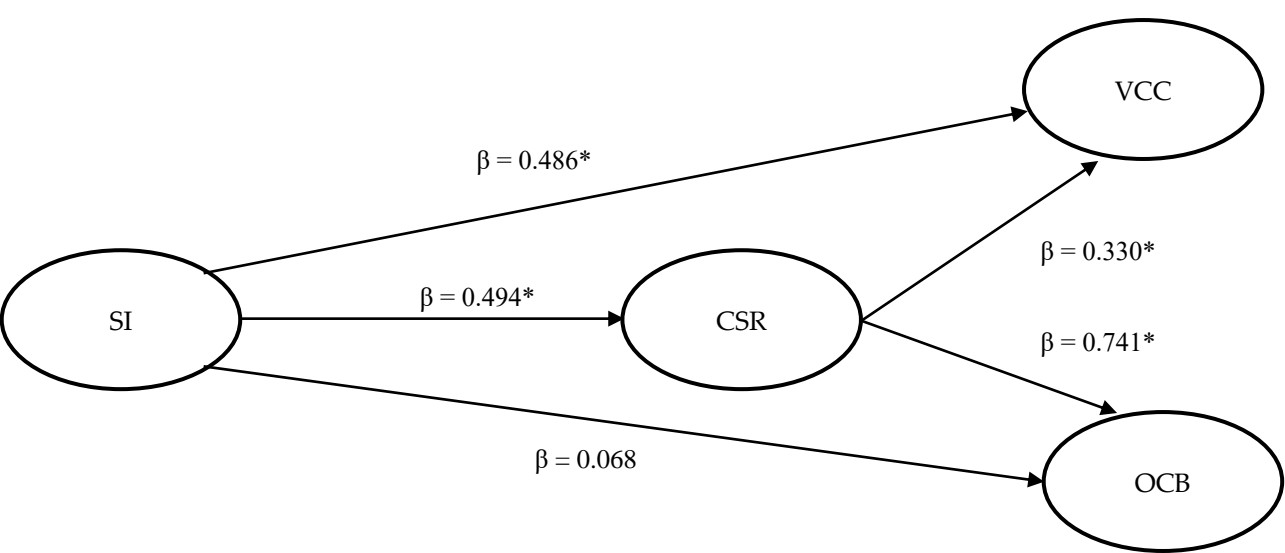

**Figure 1.** Mediating effects of attitudes toward CSR between SI, VCC and OCB. Note: The path coefficients are standardized. * $p < 0.05$.

## 5. Discussion

### 5.1. General Discussion

Drawing on stakeholder theory, this study explored the mediating effect of employee attitudes toward CSR on SI, VCC, and OCB through evaluating a series of hypotheses (H1–H5). Linear regression analyses indicate that employee perceptions of SI positively predict their attitudes toward CSR. In other words, perceptions of greater SI are correlated with superior attitudes toward CSR. This finding was consistent with the notion that employee SI perceptions in sport social enterprises may generate expectations of the organization being dedicated to altruistic actions in society [34]. Additionally, employee perceptions of



SI in the sport social enterprise were demonstrated to positively predict VCC, implying that perceptions of a higher level of SI are related to perceptions of greater VCC. This finding was consistent with the notion that VCC can be developed through cultivation of SI perceptions within a sport social enterprise [17,35]. However, employees SI perceptions did not significantly predict OCB. This finding may suggest that SI perceptions do not directly predict OCB. Rather, SI perceptions may indirectly predict OCB through the mediation mechanism of employees' attitudes toward CSR, which prompted the formulation of H4 and H5.

Linear regression analyses revealed that employee attitudes regarding the CSR adopted by their sport social enterprise can positively predict VCC and OCB. In other words, more positive attitudes toward CSR positively correlate with a greater level of VCC. This finding accords with the results of previous studies [39,40]. Furthermore, more favorable attitudes toward the CSR adopted by the sport social enterprise positively relate with a greater level of OCB. Such a finding is consistent with the findings of previous empirical studies in suggesting that OCB can be enhanced through cultivating positive employee attitudes toward CSR [41–43].

Although a thorough examination of the proposed hypotheses was ostensibly performed, the mediating mechanism of employee attitudes toward CSR on perceptions of SI, VCC and OCB warrants academic and practical attention. The analysis in this study suggested that employee attitudes toward the CSR adopted by the sport social enterprise partially mediate the relationship between perceptions of SI and VCC. This implies that SI perception can not only directly predict VCC but also indirectly predict VCC through the mediating mechanism of CSR attitudes. By contrast, employee attitudes toward the CSR adopted by the sport social enterprise fully mediate the relationship between perceptions of SI and OCB. This finding indicates that SI perceptions did not directly predict OCB; instead, they indirectly predicted OCB through a full mediating mechanism of CSR-related attitudes. The mediating role of attitudes toward CSR deepened our understanding of the mechanism between CSR and perceptions of SI, VCC, and OCB. Practitioners in sport social enterprises may consider developing internal communications with their employees about SI and CSR to generate favorable VCC and OCB outcomes.

### 5.2. Theoretical Contributions and Practical Implications

The contributions of this study are twofold. First, its theoretical contribution is providing a deeper understanding of the moderating mechanism of employee attitudes toward CSR on SI, VCC, and CSR. Second, this study made a practical and managerial contribution by examining nonprofit sport social enterprise. Although this study might be considered as at an exploratory stage, it still broadens the understanding of the moderating mechanism of employee attitudes toward CSR in the context of sport social enterprise.

### 5.3. Limitations

This study has several limitations. First, the target population in the study was employees of the sport social enterprises, which may have limited the number of research participants due to the small target population, only some of whom would have volunteered to join this study. Moreover, the construct validity was examined using EFA rather than CFA because of the small sample size. Similarly, linear regression analyses (rather than structural equation modeling) was conducted to test the proposed model, again due to limited sample size. Future studies should carefully consider the feasibility of recruiting a more appropriate number of research participants. Finally, only one sport social enterprise in Taiwan was selected as the research setting, which may limit the generalizability of the findings. Future studies can increase the generalizability of their findings by testing the proposed model in several cultural contexts.

## 6. Conclusions

Drawing on stakeholder theory, this empirical study explored the mediating mechanism of employee attitudes toward CSR (adopted by their sport social enterprise) on perceptions of SI, VCC, and OCB. Employees are critical internal stakeholders in both for-profit and nonprofit organizations. Adopting effective communications with employees on SI and CSR can benefit an organization by increasing VCC and OCB among its staff. Furthermore, a theoretical exploration of the mediating role of CSR in sport social enterprise was conducted in this study.

**Author Contributions:** Conceptualization, C.-Y.C. and Y.-L.C.; methodology, C.-Y.C. and Y.-L.C.; software, C.-Y.C. and Y.-L.C.; validation, C.-Y.C. and Y.-L.C.; formal analysis, C.-Y.C. and Y.-L.C.; investigation, C.-Y.C. and Y.-L.C.; resources, C.-S.L.; data curation, C.-Y.C. and Y.-L.C.; writing—original draft preparation, C.-Y.C. and Y.-L.C.; writing—review and editing, C.-Y.C. and Y.-L.C.; visualization, C.-Y.C.; supervision, C.-Y.C.; project administration, C.-Y.C.; funding acquisition, C.-Y.C. All authors have read and agreed to the published version of the manuscript.

**Funding:** This research received no external funding.

**Institutional Review Board Statement:** The study was conducted according to the guidelines of the Declaration of Helsinki, and approved by the Ethics Committee of National Taiwan University (protocol code NTU-REC 202107ES022).

**Informed Consent Statement:** A online informed consent statement was provided in the online survey.

**Data Availability Statement:** The dataset will be provided upon request.

**Conflicts of Interest:** The authors declare no conflict of interest.

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
