# Peer review of "Social Innovation, Employee Value Cocreation, and Organizational Citizenship Behavior in a Sport-Related Social Enterprise: Mediating Effect of Corporate Social Responsibility"

_sustainability, doi:10.3390/su132212582_

Round 1

Reviewer 1 Report

This paper focuses on examining potential impact variables that are related to corporate employees’ attitudes toward CSR. Unfortunately, at this stage, the manuscript appears to possess several fundamental flaws related to theory and hypothesis development, literature review, method and discussions. In addition, the language quality of the entire manuscript needs to be substantially improved. There are a number of grammatical mistakes, awkward sentence structures, and incorrect in-text citation formats throughout the manuscript. I provide more detailed comments below.

  1. I have major concerns about the theoretical foundation and following hypothesis development of this study. The authors provide little information/reasoning on how different layers of variables are selected. For instance, Hypothesis 1 states that social innovation can positively influence CSR attitude. However, in the literature review and hypothesis section, authors did not provide convincing rationale to support this argument. How are these two concepts connected? The following argument on the effects of social innovation on value co-creation and organizational citizenship behavior also seemed flawed. The authors failed to provide sufficient justification for the proposed relationships.
  2. The framing of the first three hypotheses reads confusing. What does “social innovation having a high perception” mean?
  3. Your first hypotheses imply causation, but with cross sectional data, you can only show evidence of correlation.
  4. Another major concern I am having of this manuscript is its literature review. Very few sport management studies on CSR have been reviewed. Authors should consider updating their relevant literature that is reflective of more recent research in the sport management field.
  5. The authors did not provide clear information on the measurement scales of each variable.  
  6. As this is a Taiwan-based study, the authors also need to discuss the generalizability and implication of the research findings in a broader context.
  7. There are numerous confusing/awkward sentence structures and grammar errors throughout the manuscript. The meaning of many sentences and passages are hard to follow. Authors should consider seeking professional language editing service to improve readability.

In summary, at present there appears to be a number of major flaws in the method, theory and framing of the paper. We wish the authors good luck in refining the manuscript as they move forward with this project.

Author Response

Authors thank and agree with reviewers comments. Please refer to the attached file about all responses.

Reviewer 2 Report

I have carefully reviewed this paper title: "A Study of Organizational Innovation, Employee Co-creation 2 of Values, and Organizational Citizenship Behavior: An Analy-3 sis of the Mediating Effects of Corporate Social Responsibility ".  

  1. Abstract:It is suggested you explain the objective of the research in the abstract. The objective of the study is not clearly explaining in the abstract. Similarly, the methodology, results, and conclusion of the study are not explaining well in the abstract.  
  2. Introduction:The introduction is not written well authors have compressed the introduction with the literature review. The introduction should be much more focused. The research objectives should be much clearer. Perhaps it could be helpful to explicitly articulate the research question. Similarly, authors need to clearly state the value-added of the paper and better discuss how this work could be worth it for both academics and practitioners. I suggest you explain the answer to mentioned four questions in your introduction part.
  1. What are the academic research questions of this study?
  2. Which theory is supporting your study? You need to find a theory which support your study and the reflection of that theory will show throughout the paper.
  3. I suggest to the authors in the last paragraph of the introduction explains the structure of the paper. 
  4. I also suggest you re-write your introduction and fallow the below mentioned papers also cite these papers into your paper.  
  5. Abbas, J., Zhang, Q., Hussain, I., Akram, S., Afaq, A., & Shad, M. A. (2020). Sustainable innovation in small medium enterprises: the impact of knowledge management on organizational innovation through a mediation analysis by using SEM approach. Sustainability12(6), 2407.
  6. Wang, M., Tang, M., Saeed, A., & Iqbal, J. (2021). How toxic workplace environment effects the employee engagement: the mediating role of organizational support and employee wellbeing. International journal of environmental research and public health18(5), 2294.
  1. Literature Review:Authors have explained their literature review in the introduction. I suggest the authors create a new heading with the title of a literature review or hypothesis development and explain the arguments in support of hypothesis development. Moreover, arguments do not flow logically, and ideas need more clarification.
  1. What theory supports your study?
  2.  It is suggested to explain the theory which supports your finding, and the reflation of this theory must show throughout the manuscript.  Below mentioned studies will help you to improve your literature review, follow the literature and cite these papers in your study.
  3. Abbas, J., Mahmood, S., Ali, H., Ali Raza, M., Ali, G., Aman, J., ... & Nurunnabi, M. (2019). The effects of corporate social responsibility practices and environmental factors through a moderating role of social media marketing on sustainable performance of business firms. Sustainability11(12), 3434.
  4. Hussain, S. T., Abbas, J., Lei, S., Jamal Haider, M., & Akram, T. (2017). Transactional leadership and organizational creativity: Examining the mediating role of knowledge sharing behavior. Cogent Business & Management, 4(1), 1361663.
  5. Asad, Ali, Jaffar Abbas, Muhammad Irfan, and Hafiz Muhammad Ali Raza. "The Impact of HPWS in Organizational Performance: A Mediating Role of Servant Leadership." Journal of Managerial Sciences 11 (2017).
  6. Wang, B., Zhao, Y., Samma, M., & Iqbal, J. (2021). Investigating the nexus between critical success factors, despotic leadership, and success of renewable energy projects. Environmental Science and Pollution Research, 1-11.
  1. Research Methodology: The research methodology of this study is not explained well. I suggest you follow the below-mentioned instructions.
  • Which research approach the authors use in this study?
  • Develop sub-heading with the title of Research approach and explain what research approach you used in this study and why you use this research approach specifically in your study.
  • How you develop your research instrument. I suggest you create a heading with the title of “Instrument development or questionnaire development” and explain the complete process of instrument development.
  • Under the heading of measurement, explain the measurement of each variable in detail.
  • It is also suggesting you create one more heading with the title of the “Respondent Summary or Demographics”. Under this heading, explain the detail about the respondents.  
  • It is suggested to you once carefully read the research methodology of below-mentioned studies it will help you to improve the study also suggest cire these papers in your methodology section. 
  • Samma M, Wang M, Zhao Y, Zhang Y. (2019). How Human Resource Management Practices Translate Into Sustainable Organizational Performance: The Mediating Role Of Product, Process And Knowledge Innovation. Psychology Research and Behavior Management, 12, 1009-1025.
  • Wang, Z., Zaman, S., uz Zaman, Q., & Amin, A. (2020). Exploring the Relationships Between a Toxic Workplace Environment, Workplace Stress, and Project Success with the Moderating Effect of Organizational Support: Empirical Evidence from Pakistan. Risk Management and Healthcare Policy, 13, 1055.
  1. Results: The results are explained well.

  1. Discussion: This section is feeble; it is not well explained. I suggest explaining the discussion part in detail and integrate it with the results and previous literature.  Below mentioned studies will help you improve the study also, suggest cites these papers in your discussion section. 

  • Asad, Ali, Jaffar Abbas, Muhammad Irfan, and Hafiz Muhammad Ali Raza. "The Impact of HPWS in Organizational Performance: A Mediating Role of Servant Leadership." Journal of Managerial Sciences11 (2017).

  1. Conclusion: The conclusion section is missing. I suggest you conclude your study and integrate your conclusion with the introduction, theory, and findings.  .  Below mentioned study will help you improve the study also, suggest cite this study in your study.
  • Abbas, J., Hussain, T., Mubeen, R., Wei, Z., & Raza, S. Social media efficacy in crisis management: Effectiveness of non-pharmaceutical interventions to manage the COVID-19 challenges. Frontiers in Psychiatry, 1732.
  1. References: It is recommended to make use of recent references to support these arguments (ideally, published during the past 5 years).

Author Response

(The authors gave the same response as above.)

Reviewer 3 Report

The paper's overall level is right: it is well written even if it is quite simple, and some important considerations are highlighted. The discussion should be rather organized around arguments avoiding simply describing details without providing much meaning. A real discussion should also link the findings of the study to theory and literature. I suggest a major revision

-In the introduction, I suggest authors must open the discussion with the importance of the innovation, and then linked it with the organization and then highlights its linkages with the OCB.For example, introductory lines may highlight the importance of innovation. See

Enhancing green product and process innovation: Towards an integrative framework of knowledge acquisition and environmental investment. Business Strategy Environment. 2020;1–13. https://doi.org/10.1002/bse.2684

-my suggestion is also to rewrite this section to answer the following questions fully: (i) Why is this topic relevant, and what is known about it? (ii) Which are the gaps you plan to address, and how do you problematize them?

If your outcome aspect is the new concept of your particular context, this should be a focus of your study and correspondingly included in the literature review. In particular, you need to show why this outcome is relevant in this study. While they are related concepts, they are different theoretically and practically. You need to focus and elaborate more on this concept and its relevance to your study area and the context. 

-Heading, 1.1, it must be moved under the heading of literature review.

-The authors did not identify other literature on the topic and explain how the study relates to this previously published research? Authors make the mistake of simply telling the reader what they found without involving the reader in the discovery process. This means that the reader is presented with statements about what the findings are, but not how the data actually support these findings or how the researchers have arrived at their interpretations (i.e., too much telling, too little showing.

-Expand the disussion in heading 1.3. Social Innovation and Corporate Social Responsibility Attitudes. Here, I suggest the authors highlight the importance of social sustainability. For example, see for the manuscript improvement.

Buyer-Driven Knowledge Transfer Activities to Enhance Organizational Sustainability of Suppliers, Sustainability 2020, 12(7), https://doi.org/10.3390/su12072993

-Especially noteworthy is that the authors state very clearly what they mean to accomplish and which other research questions they are not going to examine. The authors need to explain in sufficient detail exactly how they conducted their study (i.e., transparency of research actions), how their research approach aligned with the research question(i.e., purpose and coherence), and how their chosen methodology generated appropriate and relevant evidence vis-a-vis their research question.  The setting of their study (i.e., the research context); the sample (including sample characteristics and recruitment of the sample), the research design, and the specific application of this method in the current study.

-- The description of the chosen method can then specifically address how the method fulfils these requirements and allows the researchers to study the research question. In addition, the description of the method should discuss what information the method allows the researchers to uncover.

The main weakness of the paper is the weak analysis of the data. Although the sample is not very large with respect to the population, it is true that it is large enough to apply more complex statistical techniques that allow the author to get more relevant results.

-The section devoted to the explanation of the results suffers from the same problems revealed so far. Your storyline in the results section (and conclusion) is hard to follow. Moreover, the conclusions reached are far from what one can infer from the empirical results. The discussion should be rather organized around arguments avoiding simply describing details without providing much meaning. A real discussion should also link the findings of the study to theory and/or literature. 
Mohr, Lois A., and Deborah J. Webb. "The effects of corporate social responsibility and price on consumer responses." Journal of consumer affairs 39, no. 1 (2005): 121-147.

Turnipseed, Patricia H., and David L. Turnipseed. "Testing the proposed linkage between organizational citizenship behaviours and an innovative organizational climate." Creativity and Innovation Management 22, no. 2 (2013): 209-216.
Minor General Comments

 - The manuscript is potentially original contributive but needs a major revision.

-Some sentences from the conclusion could be moved up in the discussion section. The conclusion section must be well written and clearly explain the study findings.

- Implications for future research may also be included in the conclusion at the end. This research has article has created a lively discussion on so many issues that were hitherto unheard of and not addressed.

- Also, explain briefly what future research opportunities are.

Author Response

(The authors gave the same response as above.)

Round 2

Reviewer 1 Report

Thanks for responding to my comments in your response letter as well as in the paper itself.  I think my comments have been answered. I appreciate the effort you have put into this manuscript!

Reviewer 3 Report

I recommend that the authors improve the flow of the manuscript, ensure it follows the standard manuscript format, and recheck typos. Currently, it contains some issues, such as using incorrect forms of in-text citations, repeating the same sentences, and using inconsistent literature review arguments. It would also be good with a more extensive and convincing discussion of why not chosen mediators can be expected to be especially important. Theoretically, it is not justify without the control variables, and it would be dangerous to, Unfortunately, however, this manuscript has many weaknesses and unclear points that make it difficult to follow. Overall, this paper needs to be more or less completely rewritten and re-focused in order to merit publication in a journal. Also, I don't think the title's first paper reflects what the authors are trying to do in the paper.  Methodologically, I think the authors need to clarify their data collection and consider other control variables and cited papers are out of the sutyd unit of analysis.